# Nanoscale Hierarchical Structure of Twins in Nanograins Embedded with Twins and the Strengthening Effect

**Haochun Tang [1], Tso-Fu Mark Chang [1,\*], Yaw-Wang Chai [2], Chun-Yi Chen [1], Takashi Nagoshi [3], Daisuke Yamane [1], Hiroyuki Ito [1], Katsuyuki Machida [1], Kazuya Masu [1] and Masato Sone [1,\*]**

[1] Institute of Innovative Research, Tokyo Institute of Technology, Yokohama 226-8503, Japan
[2] School of Materials and Chemical Technology, Tokyo Institute of Technology, Yokohama 226-8503, Japan
[3] National Institute of Advanced Industrial Science and Technology, Tsukuba Ibaraki 305-8564, Japan
\* Correspondence: chang.m.aa@m.titech.ac.jp (T.-F.M.C.); msone@pi.titech.ac.jp (M.S.);
  Tel.: +81-45-924-5631 (T.-F.M.C.)

**Abstract:** Hierarchical structures of 20 nm grains embedded with twins are realized in electrodeposited Au–Cu alloys. The electrodeposition method allows refinement of the average grain size to 20 nm order, and the alloying stabilizes the nanoscale grain structure. Au–Cu alloys are face-centered cubic (FCC) metals with low stacking fault energy that favors formation of growth twins. Due to the hierarchical structure, the Hall–Petch relationship is still observed when the crystalline size (average twin space) is refined to sub 10 nm region. The yield strength reaches 1.50 GPa in an electrodeposited Au–Cu alloy composed of 16.6 ± 1.1 nm grains and the average twin spacing at 4.7 nm.

**Keywords:** nanotwin; nanograin; Au–Cu alloy; micro-compression; yield strength

---

## 1. Introduction

The usage of precious metals in micro-components of microelectromechanical system (MEMS) devices has been demonstrated to allow further enhancement in the sensitivity and miniaturization of the device [1–3]. Among the precious metals, Au is a promising material owing to its advantageous properties and process feasibility in electronic devices [4]. However, concerns regarding the structural stability of gold-based components have been noticed due to the relatively low mechanical strength. Although an improved yield strength ($\sigma_y$) of ~500 MPa [5] has been reported by refining the average grain size (*d*) to nanoscale following the Hall–Petch relationship (HP) [6–8], the strength is still low when compared with materials commonly used in electronic devices. For example, silicon materials are often applied in MEMS devices and possess fracture strength of 1–3 GPa [9]. Besides, enhancement in the strength along with the grain refinement reverses when the average grain size reaches ca. 20 nm [10–13], which is known as the inverse Hall–Petch relationship (iHP). Another strengthening utilizing the HP can be achieved through introduction of twin boundaries into the grains [14], but iHP still occurs when the average twin spacing ($\lambda$) reaches ca. 10 nm [15].

In addition to the mechanical properties, there are numerous reports on effects of nanoscale structure on fundamental properties of the material, such as, superconductivity observed in nanostructured $HgBa_2CuO_{4+y}$ [16], $La_2CuO_{4+y}$ [17], and Au–Ag [18]. The phonon density of states of Sn films are reported to be affected by the morphology and grain sizes in nanoscale [19]. Furthermore, electrodeposition is an effective method to control the structures in nanoscale [20].

Enhancement of the mechanical strength by solid solution strengthening can be achieved by alloying of the nanocrystalline Au [11–13]. The yield strength reaches 1.0 GPa in Au–Cu alloys prepared by electrodeposition and evaluated by uniaxial micro-compression tests [21,22]. The high yield strength

is a result of synergistic effects of grain boundary and solid solution strengthening mechanisms and the sample size effect [23]. On the other hand, a continuous increase in $\sigma_y$ of the electrodeposited Au–Cu alloys is observed when the grain size is lower than 10 nm, which is against the iHP reported for Au–Cu alloys when the grain size is in sub 10 nm region [10–13]. The grain sizes reported in previous works were estimated by X-ray diffraction and the Scherrer equation. Grain sizes evaluated by the Scherrer equation are recognized to be close to the real grain sizes observed by transmission electron microscopy (TEM) [24,25] in homogeneous nanocrystalline metals. However, deviations between the Scherrer equation and the TEM results could occur when there is another ordered crystalline structure in the specimen. For instance, twins in face-center cubic (fcc) metals having medium-to-low stacking fault energy ($\gamma_{sf}$) are commonly observed, such as gold [26,27] and copper [28], and electrodeposition is an effective method to cause evolution of twins [28–30]. Although there is still no report on formation of twins in Au–Cu alloys, it is necessary to investigate microstructures of the Au–Cu alloys via TEM observation to elucidate the strengthening observed in the iHP region.

Furthermore, the Au–Cu micro-pillar with high copper content shows a gradual decrease in the flow stress just after the yielding point in the stress-strain curve; while the flow stress steadily increases after the yielding for the Au–Cu micro-pillar with a low copper concentration (below 15 at.%) [22]. Such a stress drop phenomenon is rarely reported in nanocrystalline face-centered cubic (fcc) metals and should be clarified.

In this work, formation of twins in the electrodeposited Au–Cu alloys is verified to disclose the continuous strengthening observed in the iHP region. In addition, microstructures of the Au–Cu micro-pillar are evaluated to understand the stress drop observed in the stress–strain curve.

## 2. Materials and Methods

Au–Cu alloy films were electrodeposited with an electrolyte containing $X_3Au(SO_3)_2$ (X = Na, K) and $CuSO_4$. Details of the electrodeposition procedures are reported in previous studies [21,22]. The chemical composition and crystal structure were characterized by energy-dispersive spectroscopy in a scanning electron microscope (SEM, Hitachi SU4300SE, Tokyo, Japan) and X-ray diffraction (XRD, Rigaku Ultima IV, Tokyo, Japan). For characterization of the mechanical property and in consideration of the sample size effect for MEMS applications, micro-pillars fabricated from the Au–Cu alloy films were prepared. The Au–Cu alloy film electrodeposited specimens were first thinned down to less than 100 μm by mechanical polishing and cut into semicircle disk shapes by a mechanical punch machine. Then micro-pillars with dimensions of $15 \times 15 \times 30$ μm$^3$ were fabricated using a focus ion beam (FIB, Hitachi FB2100, Tokyo, Japan). Mechanical properties of the Au–Cu alloy micro-pillars were evaluated by micro-compression tests with a displacement-control mode, and the strain rate was $5 \times 10^{-3}$ s$^{-1}$. More details of the micro-mechanical testing equipment are described in a previous study [31]. Microstructures of the as-deposited Au–Cu alloys and the deformed micro-pillars were observed using a scanning TEM (STEM, JEOL JEM-2100F, Tokyo, Japan) equipped with a high-resolution TEM (HRTEM) operated at 200 kV. Specimens used in the STEM and TEM were prepared by MultiBeam SEM-FIB (JEOL JIB-4500, Tokyo, Japan). For the deformed specimens, the milling direction of the Ga ion beam in the FIB was parallel to the compression direction.

## 3. Results and Discussion

Electrodeposited Au–Cu alloys incorporated with nanotwins were confirmed by STEM and HRTEM observation. Figure 1a,b shows the STEM images of the $Au_{85}Cu_{15}$ (15 at.% Cu) and $Au_{68}Cu_{32}$ (32 at.% Cu) alloys, respectively. Individual nanoscale crystal grains and the boundaries could be distinguished from contrasts of the patterns. The average grain sizes were 25.6 ± 4.1 and 16.6 ± 1.1 nm for the $Au_{85}Cu_{15}$ and $Au_{68}Cu_{32}$ alloys, respectively. Nanotwins were observed in the STEM images as indicated by the arrows in Figure 1a,b. Figure 1c shows XRD patterns of the as-electrodeposited $Au_{85}Cu_{15}$ and $Au_{68}Cu_{32}$ alloys. No diffraction peaks from other ordered structure (i.e., L1$_2$ Au$_3$Cu or L1$_0$ AuCu) were observed except the fcc diffraction peaks, indicating complete solid solution in

the electrodeposited Au–Cu alloys. The average sizes of the ordered crystalline estimated by the XRD results and the Scherrer equation were 7.8 and 4.7 nm for the $Au_{85}Cu_{15}$ and $Au_{68}Cu_{32}$ alloys, respectively. The grain sizes observed in STEM (*d*) were much larger than the average sizes from the Scherrer equation, which implied the average sizes were very likely to be average spacing of the nanotwins ($\lambda$). Figure 1d is a representative HRTEM image of the $Au_{85}Cu_{15}$ alloy, which shows a ~30 nm grain containing a ~8 nm wide single band. The electron diffraction patterns converted by fast Fourier transform (FFT) confirmed the nanotwin structure, and the twin is symmetrical to the matrix with the twin boundary (TB) (111) plane. The grain can be divided into three individual bands by the parallel TBs and the widths are all about 10 nm, which is very close to the $\lambda$ estimated by the Scherrer equation. On the other hand, grains containing only one TB were also observed. As shown in Figure 1e, the TB located in the middle of the grain separates the grain into two equal parts. Illustration of grains divided by one and two TBs is shown in Figure 1f.

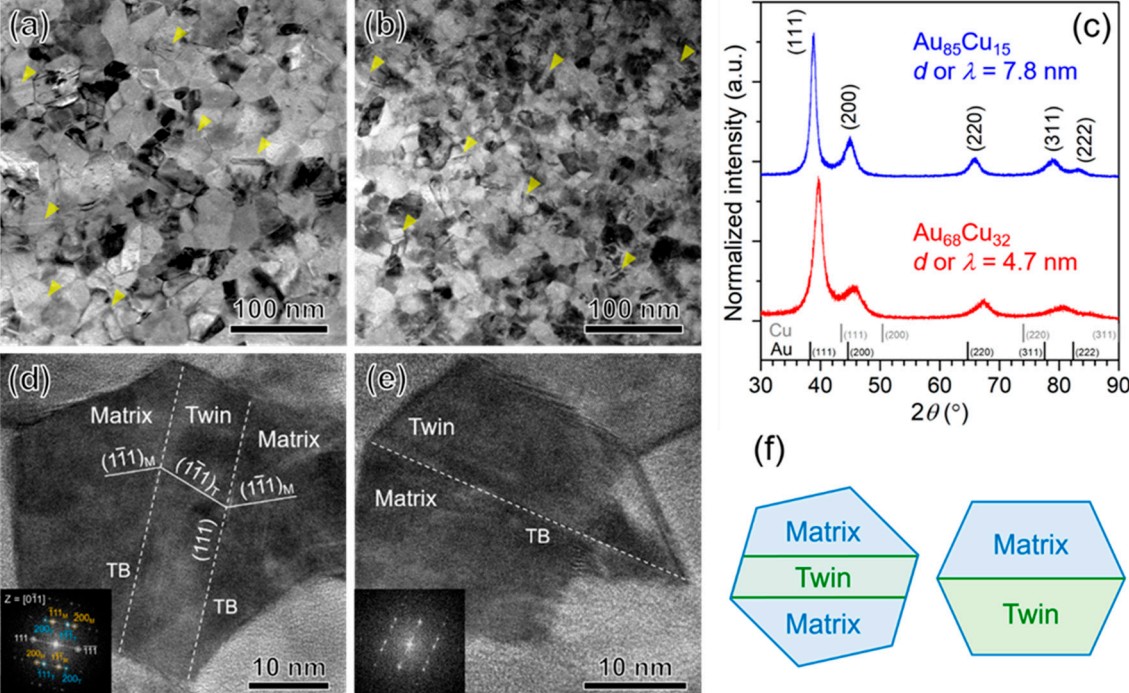

**Figure 1.** (**a**,**b**) Bright-field scanning transmission electron microscopy (STEM) images of as-electrodeposited $Au_{85}Cu_{15}$ and $Au_{68}Cu_{32}$ alloys. The arrows indicate the nanotwins inside the nanograins. (**c**) XRD patterns of $Au_{85}Cu_{15}$ and $Au_{68}Cu_{32}$ alloys. The vertical bars at bottom indicate the diffraction peaks of pure Au and Cu. (**d**,**e**) Two representative high-resolution transmission electron microscopy (HRTEM) images taken from the $Au_{85}Cu_{15}$ alloy. Zone axis: [$0\bar{1}1$]. The vertical bars at bottom indicate the diffraction peaks of pure Au and Cu. (**f**) Illustration of two types of the nanotwin in a nanograin.

For alloy electrodeposition, the applied current density plays an important role in controlling the grain size and composition. In the case of Au–Cu alloys, the Cu concentration is increased by applying a higher cathodic current density due to the difference in standard reduction potential between Au and Cu ions [11,13]. Meanwhile, the higher current density can promote the nucleation rate resulting in finer grains in electrodeposits [32]. The twin evolution is attributed to the lowered $\gamma_{sf}$ by alloying two fcc metals already with relatively low $\gamma_{sf}$. A strong decrease in the $\gamma_{sf}$ as a result of alloying was experimentally examined and revealed to have a semi-log relationship in most of fcc-based alloys (i.e., Ag, Cu, Ni) as expressed in the following [33,34]:

$$\ln \frac{\gamma_{sf}}{\gamma_0} = k_\gamma \left( \frac{x}{1+x} \right)^2, \tag{1}$$

where $\gamma_0$ is the stacking fault energy of the solvent metal. $k_\gamma$ is a material constant. $x$ is the expression of $c/c^*$, where $c$ is the solute concentration, and $c^*$ is the solubility limit. For example, the stacking fault energy of pure Cu reduces from ~70 mJ/m$^2$ to a value lower than 10 mJ/m$^2$ when forming Cu-based alloys [33]. Wu et al. reported the formation of a nanotwinned structure in electrodeposited Ni–80Co alloys with average grain size of ~30 nm, and the growth twins were reported to be affected by $\gamma_{sf}$ of the alloy [29]. Lucadamo et al. also observed the twinning features in electrodeposited Ni–Mn alloys but with coarser grains of ~200 nm [30].

Micro-mechanical properties of the electrodeposited Au–Cu alloys were revealed by micro-compression tests. Figure 2a–d shows SEM images of the as-fabricated Au$_{85}$Cu$_{15}$ and Au$_{68}$Cu$_{32}$ micro-pillars and after compression with 12%–14% plastic strain. Similar barrel-shape deformations were observed in both micro-pillars, which were typical deformation behaviors for polycrystalline metallic materials. The engineering stress–strain curves obtained from the compression tests are shown in Figure 2e. The $\sigma_y$'s of the Au$_{85}$Cu$_{15}$ and Au$_{68}$Cu$_{32}$ micro-pillars were 0.95 and 1.16 GPa, respectively. After the yielding point, the Au$_{85}$Cu$_{15}$ pillar exhibited a steady increase in the flow stress during the plastic deformation until unloading. For the Au$_{68}$Cu$_{32}$ pillar, the flow stress declined in the early stage of the plastic deformation for strain of ~2%. After that, the flow stress steadily increased similar to that of the Au$_{85}$Cu$_{15}$ pillar. It should be noticed that there is still no report on the stress drop for pure polycrystalline fcc micro-specimens.

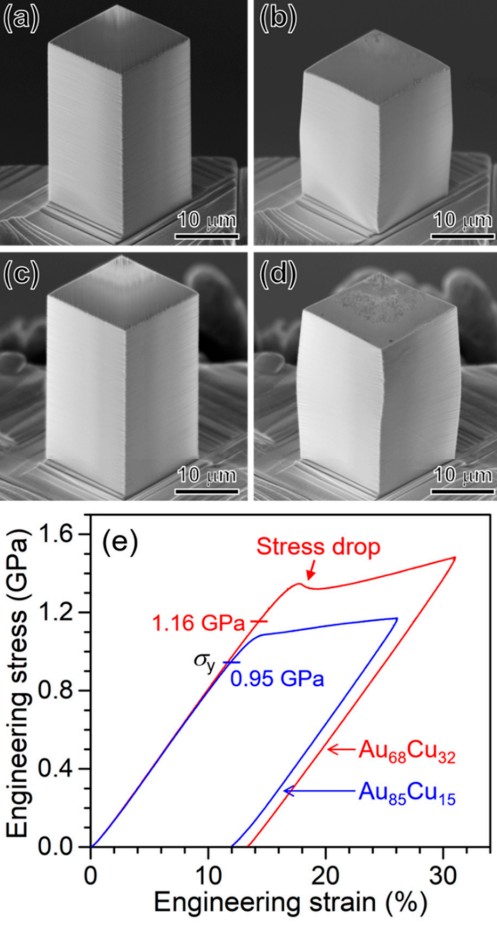

**Figure 2.** SEM images of (**a**,**b**) Au$_{85}$Cu$_{15}$ and (**c**,**d**) Au$_{68}$Cu$_{32}$ micro-pillars (**a**,**c**) before and (**b**,**d**) after the compression with 12%–14% strain. (**e**) Engineering stress–strain curves obtained from the micro-compression tests.

To understand the stress drop observed in the stress–strain curves, microstructures of the deformed micro-pillars were further investigated by the STEM and HRTEM. Figure 3a shows a STEM image of the $Au_{68}Cu_{32}$ alloy after compression of 13.8% plastic strain. Similar to the as-electrodeposited alloys shown in Figure 1a,b, conspicuous nanotwins were observed inside the nanograins. In addition to the growth twins, deformation twins inside highly deformed grains were observed as shown in Figure 3b. In the image, a deformation TB next to a growth TB located at the left side of the grain was observed, and the deformation TB was obstructed in the middle of the grain. Another deformation TB could be observed at the right side of the grain. A magnified inverse fast Fourier transform (IFFT) image shown in Figure 3c reveals the extremely complex interaction between the deformation twin and dislocation, which forces the twinning to be interrupted inside the grain.

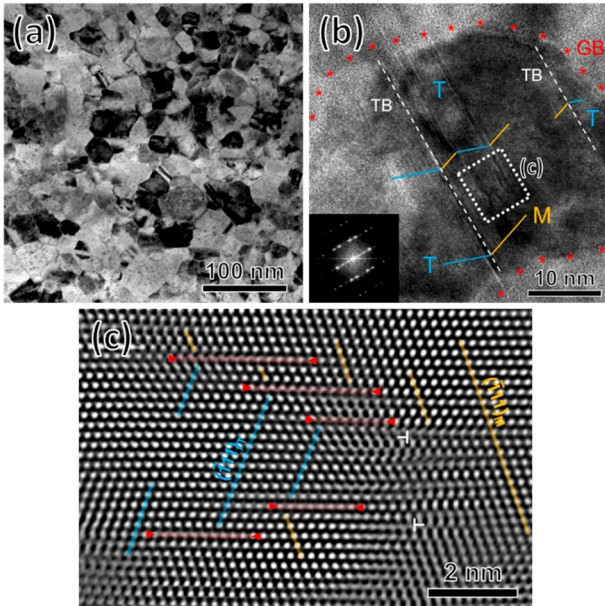

**Figure 3.** (**a**) A STEM image of the $Au_{68}Cu_{32}$ micro-pillar after ~13.8% compressive strain. (**b**) An HRTEM image of a highly deformed grain showing deformation twin. T: twin, M: matrix, Zone axis: $[0\bar{1}1]$, and (**c**) a magnified inverse fast Fourier transform (IFFT) image showing the deformation twin.

Deformation twinning is one of major deformation mechanisms not only in fcc metals with low $\gamma_{sf}$, but also in nanocrystalline fcc metals with high $\gamma_{sf}$ if deformed under extreme conditions [35,36]. Several deformation twinning mechanisms are proposed and observed in nanocrystalline fcc metals, i.e., the random activation of partials mechanism [37], the dislocation rebound mechanism [38], or the partial emissions from grain boundary [35,38]. When a twin structure initiates from the grain boundary and terminates inside a grain, it can only be formed by the partial emissions from grain boundary. Zhu et al. [39] observed similar results in nanocrystalline Ni and proposed the relative mechanisms for Shockley twinning partials to multiply at grain boundary (GB). Furthermore, the $\gamma_{sf}$ of the fcc metals is usually reduced by alloying, especially for the Au–Cu alloys. Therefore, the reduction in $\gamma_{sf}$ can change the energy path (i.e., general planar fault energy [40]) and, thus, facilitates the deformation twinning under the applied stress. The stress drop observed in the $Au_{68}Cu_{32}$ alloy pillar is reasonably considered to be the lowered energy requirement for Shockley twinning partials threading into grains to form deformation twins.

Mechanical strengths of polycrystalline metals are often affected by multiple strengthening mechanisms taking place simultaneously. In the present case of electrodeposited Au–Cu alloys, the obtained $\sigma_y$ are considered to be the synergistic effects of grain boundary strengthening, twin boundary

strengthening, and solid solution strengthening. The effect of grain size on the strength is known to be the Hall–Petch relationship [6,7]:

$$\sigma_{\text{gb}} = \sigma_0 + k_{\text{HP}} \cdot d^{-1/2}, \tag{2}$$

where $\sigma_{\text{gb}}$ is the strength contributed from GB, $\sigma_0$ is the friction resistance for dislocation movement within the polycrystalline grains, $k_{\text{HP}}$ is the Hall–Petch coefficient, and $d$ is the grain size. The twin boundary could form barriers to the dislocation motion similar to the grain boundary. Lu et al. [15,28,41] reported that the average twin width ($\lambda$) and strength of the specimen follows a Hall–Petch relationship-like behavior in the columnar-grained Cu with high density nanotwins perpendicular to the growth direction. On the other hand, the nanotwin in columnar grain structure is different from the ones present Au–Cu alloys. The Au–Cu alloys evaluated in this study were composed of isotropic grains of much smaller grain size, and because of the ~20 nm average grain size, each grain could accommodate a low number (mostly one and two in this study) of the twin boundaries and resulted a sub 10 nm average twin width. Figure 4a shows the Hall–Petch plot for Au–Cu alloys including the results of the present study and literatures evaluated by Vickers hardness tests [10,12,13]. $\sigma_y$ of the Au–Cu alloys increased from 0.90 to 1.50 GPa, when the $\lambda$ decreased from 4.7 to 9.1 nm.

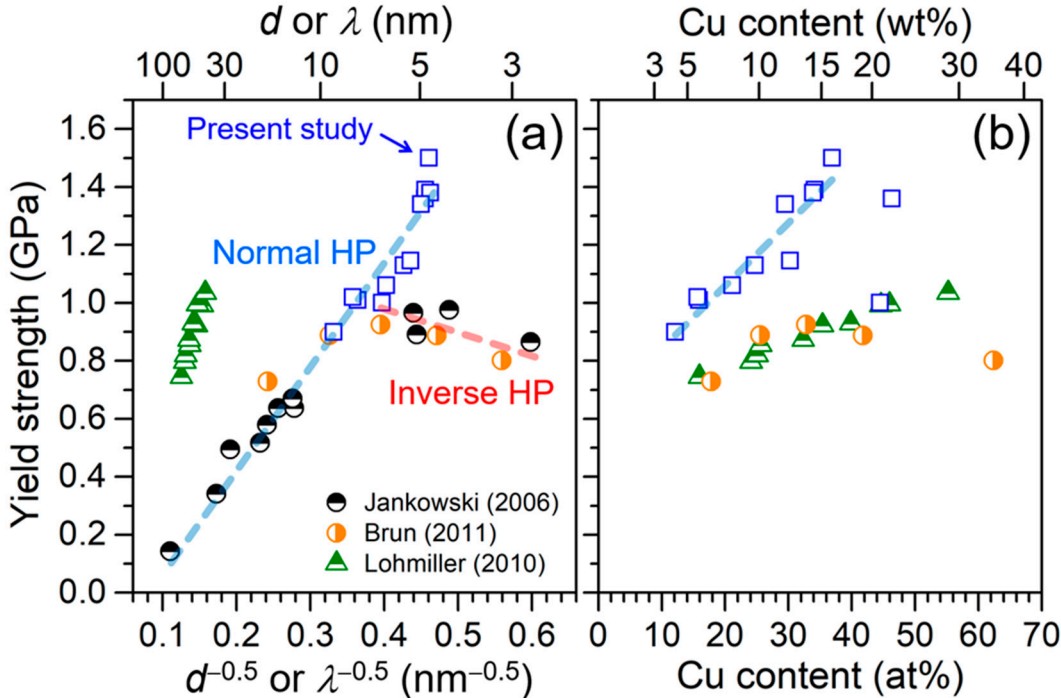

**Figure 4.** (**a**) Hall–Petch plot of Au–Cu alloys. The yield strengths in the literatures were converted from Vickers hardness. (**b**) A plot of yield strengths as a function of Cu content.

For the solid solution strengthening, the classical theories are well established in coarse-grained alloys such as Fleischer model [42] and Labusch theory [43]. Rupert and Schuh et al. further proposed enhanced models for nanocrystalline fcc alloys, in which the $\sigma_y$ and the strength contributed from nanocrystalline solid solution ($\Delta\sigma_{nc,SS}$) are expressed by [44,45]:

$$\sigma_y = A \cdot E, \tag{3}$$

$$\Delta\sigma_{\text{ns,SS}} = A \cdot \left(\frac{\partial E}{\partial c}\right) \cdot C, \tag{4}$$

where $A$ is a fitting constant having a function of the applied strain rate and grain size, $E$ is elastic modulus of the alloy, $c$ is composition in at.%. Equations (3) and (4) suggested that the strength in

nanocrystalline alloys is not only dominated by the grain size but also affected by the elastic modulus and composition. The copper concentration of the Au–Cu alloys prepared in this study ranged from 12.1 to 46.4 at.%. Here, we assume two conditions to approach the constant *A*: (i) grain sizes in all Au–Cu alloys are similar and (ii) *E* follows a linear fashion with alloy composition and ranges between the elastic modulus of Au (74 GPa) and Cu (117 GPa). By doing the assumptions, the fitting constant *A* is equal to 0.0375, which is somewhat larger than the value reported for nanocrystalline Cu alloys (0.024) [45]. Nevertheless, this modified model for nanocrystalline alloys is in line with our experimental results as shown in Figure 4b.

Au–Cu alloys prepared in this study were confirmed to have ~20 nm as the average grain size and sub 10 nm as the average twin spacing. Both values were still in the HP region and close to the critical value for occurrence of the iHP, which demonstrated thorough utilization of the HP in strengthening of Au–Cu alloys. Due to this, an ultrahigh yield strength of 1.5 GPa was obtained.

## 4. Conclusions

A hierarchical nanostructure of nanocrystalline Au–Cu alloys containing nanotwins was produced by electrodeposition from sulfite-based electrolyte. Microstructure investigation revealed that average grain sizes of the alloys were about 20 nm, and twin boundaries were observed in the nanograins. Due to the fine grain size, average spacings of the twins were all less than 10 nm, and this confirmed continuous strengthening was observed when the average twin spacing is thinned downed to sub 10 nm region. By making a hierarchical structure of twinned nanograins having the size in the HP region but close to the iHP region, a high yield strength of 1.5 GPa was obtained. In addition, the stress drop observed in the stress–strain curve was caused by evolution of the deformation twins, and the deformation twins were formed because of the reduced stacking fault energy in the Au–Cu alloys.

**Author Contributions:** Conceptualization, H.T., T.-F.M.C., and M.S.; data curation, H.T. and T.-F.M.C.; validation, H.T., C.-Y.C., H.I., and D.Y.; formal analysis, H.T. and T.N.; investigation, H.T. and Y.-W.C.; resources, M.S.; writing—original draft preparation, H.T.; writing—review and editing, T.-F.M.C.; visualization, H.T.; supervision, T.-F.M.C. and M.S.; project administration, K.M. (Katsuyuki Machida) and M.S.; funding acquisition, K.M. (Kazuya Masu) and M.S.

**Funding:** This work was supported by JST CREST grant number JPMJCR1433, Japan and the Grant-in-Aid for Scientific Research (S) (JSPS KAKENHI grant number 26220907).

**Acknowledgments:** The authors thank Suzukakedai Materials Analysis Division, Technical Department, Tokyo Institute of Technology, for TEM specimen preparation.

**Conflicts of Interest:** The authors declare no conflict of interest.

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
