# Peer review of "Nanoscale Hierarchical Structure of Twins in Nanograins Embedded with Twins and the Strengthening Effect"

_metals, doi:10.3390/met9090987_

Round 1

Reviewer 1 Report

The paper “Nanoscale Hierarchical Structure of Twins in Nanograins Embedded with Twins and the Strengthening Effect” by Haochun Tang et al. reports interesting experimental results on nanoscale hierarchical structures in grains embedded with twins in electrodeposited Au-Cu alloys.

The paper is suitable for publication with minor changes.

In the paragraph discussing micro-compression tests reported in figure 2 (lines 117-139) and in the paragraph discussing micro-compression tests reported in figure 3 (lines 146-153) the authors should expand the description the experimental methods adding details of the used procedure for the reader.

In the conclusion the authors should add a sentence on the relevance of their results in other fields as in nanostructured oxides where nanoscale grains, twins, are controlling quantum functionalities in these materials [1]. Indeed, in these materials nanoscale imaging [2, 3] has shown textures similar to the Au-Cu alloys presented here. These results represent a new frontier in quantum materials made of nanoscale particles which can be controlled by engineering the elastic strain [4]

1. Nature, 2015, 525(7569), 359 https://doi.org/10.1038/nature14987 2. Condens. Matter 2019, 4(1), 32. https://doi.org/10.3390/condmat4010032 3. Condens. Matter 2019, 4(3), 77.  https://doi.org/10.3390/condmat4030077 4. The strain of CuO2 lattice: the second variable for the phase diagram of cuprate perovskites. Journal of Physics A: Mathematical and General, 2003, 36(35), 9133.

Author Response

Answers to Reviewer #1' comments

Reviewer 2 Report

This is a well written and documented manuscript.

A comparison with AuCu alloy produced via a method besides electrodeposition would have made for a more compelling analysis against the effect of grain size and reinforced the conclusion. However references to literature support the results adequately for publication.

Author Response

Reply to Reviewer #2

This is a well written and documented manuscript. A comparison with AuCu alloy produced via a method besides electrodeposition would have made for a more compelling analysis against the effect of grain size and reinforced the conclusion. However references to literature support the results adequately for publication.

Response:

Thank you very much for the positive comment. We agree that comparison with AuCu alloys prepared by other method could be interesting. However, reporting the nanoscale hierarchical structure is the main focus in this study. Influences of the fabrication method could be reported in another study.

Reviewer 3 Report

I have read the paper of Tang et al. with interest. It is a rounded and well executed study of nanostructures of electrodeposited Au-Cu alloys, with clear prescriptions and descriptions for the follow-up works. The paper is organized well, and fits in the scope of the journal, as submitted. Moreover, the estimated (record) yield strengths of the fabricated structures recommend this paper for publication in MDPI Metals.

One critical remark I have is that authors did not attempt to reach beyond the narrow subject of the study. In my opinion, a reflection on other related fields of study, in introduction or conclusions, would be welcome. I point out that electrocrystallization of metals into nanostructures of various shapes was pioneered in J. Am. Chem. Soc. 126, 2317 (2004), that quality of nanograins and resulting physical behavior is of great interest for e.g. Sn films (Phys. Rev. B 95, 155413 (2017)), and that recently high-temperature superconductivity was seen in crystalline Au-Ag assemblies (https://arxiv.org/abs/1807.08572v2) and created an enormous buzz in the scientific community.

With few lines added to accommodate this remark, I can recommend the paper for publication.

Author Response

Answers to Reviewer #3' comments
